# A Novel Policy Alignment and Enhancement Process to Improve Sustainment of School-Based Physical Activity Programming

**DOI:** 10.3390/ijerph20031791

**Published:** 2023-01-18

**Authors:** Penelope J. Friday, Lexie R. Beemer, Diane Martindale, Amy Wassmann, Andria B. Eisman, Thomas Templin, Ronald F. Zernicke, Lynn Malinoff, Anna Schwartz, Tiwaloluwa A. Ajibewa, Michele W. Marenus, Rebecca E. Hasson

**Affiliations:** 1School of Kinesiology, University of Michigan, Ann Arbor, MI 48109, USA; 2Birch Run Area Schools, Birch Run, MI 48415, USA; 3Saginaw Intermediate School District, Saginaw, MI 48603, USA; 4Division of Kinesiology, Health and Sport Studies, College of Education, Wayne State University, Detroit, MI 48202, USA; 5Center for Health and Community Impact, College of Education, Wayne State University, Detroit, MI 48202, USA; 6Departments of Orthopaedic Surgery and Biomedical Engineering, University of Michigan, Ann Arbor, MI 48109, USA; 7Institute for the Study of Children, Families, and Communities, Eastern Michigan University, Ypsilanti, MI 48197, USA

**Keywords:** implementation science, classroom-based physical activity interventions, exploration, preparation, implementation, sustainment framework

## Abstract

The purpose of the current study was twofold: (1) to evaluate the strength and comprehensiveness of district wellness policies in one central Michigan intermediate school district (ISD; 16 districts), and (2) to pilot a novel policy alignment and enhancement process in one district within the ISD to improve sustainment of district-wide physical activity (PA) programming. Policy evaluation and alignment were determined using WellSAT 3.0. The Exploration, Preparation, Implementation, Sustainment (EPIS) framework was used to guide a seven-step policy alignment and enhancement process. Initial evaluation of the PA policy for the ISD revealed a strength score of 19/100 (i.e., included weak and non-specific language) and 31/100 for comprehensiveness (i.e., mentioned few components of the Comprehensive School Physical Activity Program). For the pilot school district, initial strength scores were 19/100 and 38/100 for comprehensiveness (exploration). An alignment of the tailored PA policy with current practices resulted in a 100% increase in strength (score of 38/100), and 132% increase in comprehensiveness (score of 88/100; preparation). However, district administrators encountered barriers to adopting the tailored policy and subsequently integrated the PA requirements into their curriculum guide and school improvement plan (implementation and sustainment). Future research should examine the effectiveness of our EPIS-informed policy evaluation, alignment, and enhancement process to promote widespread increases in student PA.

## 1. Introduction

Physical activity (PA) is a crucial element of a child’s health, well-being, and development [1]. Recognizing the important role that schools play in promoting physical activity, the United States (US) Congress passed “The Child Nutrition and WIC Reauthorization Act” in 2004 and the “Healthy, Hunger Free Kids Act of 2010” mandating the establishment of district school wellness policies [2]. These policies are written documents that guide a school district’s efforts to create supportive physical activity and nutrition environments, and are a requirement for any district participating in the federally funded School Lunch and School Breakfast Program [3]. In 2015, the reauthorization of the Elementary and Secondary Education Act, known as the “Every Student Succeeds Act”, provided increased funding to states and school districts for health education and physical education programs [4]. Despite these legislative wins, however, 76% of US children ages 6–15 years are still not achieving the daily recommendation of 60 min of physical activity [5].

While virtually all district wellness policies have nutrition requirements, less than 4% of K-12 schools across the US require daily physical education (PE) and PA equivalents [6]. That is due in part to a lack of state-wide emphasis on PA promotion [7]. Moreover, only 22 states require schools to allot a specific amount of time for PE [8]. When states do have PA policies in place, they often lack strong and comprehensive language. Strong policy language refers to clear and specific statements that require action and include accountability measures [9]. Comprehensive policy language is defined as the extent to which recommended content areas are covered in the policy [10], such as the components of the Comprehensive School Physical Activity Program (CSPAP) framework [11]. These components include classroom PA, PE, before- and after-school PA, and family and community engagement. Policies that are vague and limited in scope are at the discretion of school staff to interpret and can thus result in subpar implementation [7].

Yet, even if policies are strong and comprehensive, putting them into place without aligning them within the school or district context will likely result in poor implementation. Many school districts across the US use model policy templates provided by state or federal agencies to develop their district wellness policies [12]. These model policy templates are often written by policy agencies or government officials and are disseminated state-wide for school districts to copy [13]. Although model policy templates are readily available, their alignment with school and district contexts varies widely [14]. In contrast, locally developed wellness policies are written to reflect the opinions, expertise, and goals of school district staff and leadership, which can promote greater alignment with existing programs and result in more successful implementation. Smith et al. reviewed and evaluated 20 West Virginian school districts’ wellness policies, 10 of which were locally developed and 10 of which were model policies [12]. The locally developed policies were significantly stronger and more comprehensive than the model policies. Most notably, the largest and most significant gap between local and model policies occurred within the Physical Education Physical Activity (PEPA) section, with the locally developed policies scoring higher in strength and comprehensiveness compared to the model policies [12].

Implementation science frameworks can guide policy development processes to support implementation success and sustainment of district-wide PA practices. The Exploration, Preparation, Implementation, Sustainment (EPIS) framework is an evidence-based, multiphase framework that has been used to understand the implementation process of evidence-based practices [15,16]. This four-phase model helps to identify stakeholder needs and evidence-based practices that can meet that need (exploration). This framework also guides the assessment of potential implementation barriers and facilitators and development of supports to overcome these barriers (preparation). Finally, EPIS monitors the implementation process (implementation) and guides the creation of accountability measures to sustain programming (sustainment). By using a process framework, such as EPIS, a systematic and evidence-based structure to guide local policy development can be established. More importantly, the use of implementation science frameworks can be the link to align policy processes across different contexts.

### Purpose

The purpose of the current study was twofold: (1) to evaluate the strength and comprehensiveness of district wellness policies in one central Michigan intermediate school district (ISD; 16 districts), and (2) to pilot a novel policy alignment and enhancement process, guided by EPIS in one district within the ISD to improve sustainment of district-wide PA programming. By co-developing a systematic process to evaluate, align, and enhance district wellness policies with local school leadership, the possibility of school-based PA programming sustainment can be substantially increased.

## 2. Materials and Methods

### 2.1. Setting and Participants

One ISD located within a low-resource, low-active county was recruited for policy evaluation. The ISD consisted of 25,784 K-12 students (48% female, 55% White, and 59% free and reduced lunch (FRL)). This county was selected for its low ranking in overall health outcomes and health behaviors that included PA (77th out of 83 counties [17]). Only 71% of the county residents reported having access to gyms, parks, and sports programs [17]. Consequently, limited PA among children and youth was identified as a significant need in this county and ISD.

The pilot school district consisted of 1847 students (50% female, 90% White, 55% FRL eligible) [18]. The pilot school district was selected based on their elementary school implementing the classroom-based PA program, Interrupting Prolonged sitting with ACTivity (InPACT) [19]. The district superintendent was approached and asked to participate in the policy evaluation, alignment, and enhancement process.

### 2.2. Policy Evaluation and Enhancement Process

Figure 1 displays the seven-step process for policy evaluation, alignment, and enhancement that was developed using the evidence-based EPIS framework phases [15,16].

### 2.3. Exploration

The exploration phase was characterized by the identification of emergent or existing needs and included Steps 1 and 2 of the policy evaluation, alignment, and enhancement process.

#### 2.3.1. Step 1: Policy Evaluation

Wellness policies for the 16 districts within the ISD were located by the regional school health coordinator and research staff. The policies were downloaded from the school district’s website or obtained from the district food services coordinator.

The Wellness School Assessment Tool 3.0 (WellSAT 3.0) [20], developed by the Yale Rudd Center for Food Policy and Obesity, was used to evaluate the strength and comprehensiveness of each district’s wellness policy. WellSAT 3.0 includes six sections, and we focused on the PEPA section for the purposes of this study. The 16 items within the PEPA section focused on time per week of PE instruction, PE substitution and exemptions, recess, community engagement of PA, the promotion of PA throughout the school day, and active transportation [20] (see Table 1). Each item was coded using a three-point scale from 0–2 points. If the item scored a “0”, then there was no mention of that item within the policy. If a score of “1” was assigned to the item, then vague or weak language was used to mention the item. If a score of “2” was assigned, then the policy item was addressed with specific strategies, met best-practice criteria, and strong language was used.

Comprehensiveness was calculated by counting the number of items in the PEPA section rated as “1” or “2,” dividing this number by the number of policy items in the section and multiplying this number by 100. Strength was calculated by counting the number of items in the section rated as “2,” dividing that number by the number of policy items in the section, and multiplying that number by 100 [20]. The PEPA section was given a unique score for both comprehensiveness and strength, ranging from 0–100, with lower scores indicating less content and weaker language, and higher scores representing more content and use of specific and directive language [20].

Four members of the research staff (i.e., M.W.M., L.R.B., A.S., and T.A.A.) were trained to use the validated WellSAT 3.0 by the Indiana Department of Health Childhood Obesity Prevention Coordinator. Research staff independently completed three practice review rounds using three wellness policies from districts in other ISDs. Each research staff then rated a random selection of the ISD wellness policies, such that each wellness policy had three independent ratings from three research staff. Next, the research staff met to compare the independent scores of their evaluations to come to a consensus on scores that differed and review any clarifications using the resources on the WellSAT 3.0 website. The interrater-reliability was 0.86.

#### 2.3.2. Step 2: District Self-Evaluation

The research team built on the WellSAT 3.0 analysis using qualitative methods to deepen understanding of PA policy and program misalignment within the pilot district. The research team met virtually with school district leadership on a thirty-minute video conferencing call to complete an online survey using Qualtrics™ software, which included questions clarifying if PA opportunities were being offered that were not reflected in the current wellness policy (i.e., the PEPA policy item received a WellSAT score of “0”). Open-ended questions were developed to assess: (1) “How much time per week, in MINUTES, of PE instruction do elementary, middle, and high school students receive?” (2) “What, if any, are the qualifications and training for PE teachers for grades K-12?” (3) “What, if any, are the PE exemption and substitution requirements for all students?” (4) “How many MINUTES of recess are provided to all elementary schools?” (5) “What, if any, PA breaks are offered to K-12 students?” (6) “Do Safe Routes to School plans or programs exist for the school district?” The research team was available to answer any clarifying questions.

### 2.4. Preparation

The preparation phase was characterized by the assessment of potential implementation barriers and facilitators and development of supports to overcome these barriers and included steps 3–5 of the policy evaluation, alignment, and enhancement process.

#### 2.4.1. Step 3: Tailored Policy Language

After the Qualtrics™ survey was completed, the research team used the responses to add strong and comprehensive language to the policy to reflect the current PA practices within the district, with the purpose of enhancing the wellness policy and practice alignment. The tailored policy was then rescored and enhancements in strength and comprehensiveness were recorded.

#### 2.4.2. Step 4: District Workshop

The scores of the original and tailored policies were presented to the key district partners at an in-person school district workshop. After reviewing the scores, qualitative methods to deepen understanding of school administrators’ perceptions of the tailored PA policy language were collected via a Strengths, Weaknesses, Opportunities, and Threats (SWOT) analysis [21]. Each district partner documented their perception of the: (1) strengths of the tailored policy, (2) weaknesses of the tailored policy, (3) opportunities an updated wellness policy could bring to the district and their students, and (4) threats to implementing the tailored wellness policy. After the individuals completed the SWOT analysis independently, they were asked to form groups of 2–3 to discuss their responses for 10 min after which the district partners engaged in large group discussion for 20 min. The small groups were asked to share their thoughts back to the larger group. The responses of the district partners in the large group discussion were documented by the research team on a SWOT analysis form.

#### 2.4.3. Step 5: Development of Updated Policy and/or Alternate Sustainment Strategy

At the conclusion of the workshop, district leadership was asked to continue discussion regarding the feasibility of adopting the tailored policy language. If the new policy was deemed feasible, district leadership was encouraged by research staff to move forward with policy approval and implementation. If the new policy was deemed unfeasible, then district leadership was encouraged to identify alternative strategies for sustaining the implementation of current PA programming within the school.

### 2.5. Implementation

The implementation phase was characterized by initiating policy implementation and monitoring the implementation process and included step 6 of the policy evaluation, alignment, and enhancement process.

#### Step 6: Policy/Strategy Implementation

District partners were asked to construct a timeline for: (1) obtaining appropriate approvals for the tailored policy and subsequent policy implementation, or (2) determining and implementing alternative strategies for PA program sustainment.

### 2.6. Sustainment

The sustainment phase was characterized by developing a process and supports for continued implementation and included step 7 of the policy evaluation, alignment, and enhancement process.

#### Step 7: Policy/Strategy Evaluation

District partners were encouraged to develop accountability measures for monitoring sustained implementation of the updated PA policy or alternative strategy. Accountability measures for continued improvement were independently developed by school district leadership.

## 3. Results

### 3.1. Exploration

#### 3.1.1. Wellness Policy Scores (Step 1)

Policy evaluation was completed in December 2020. Table 1 displays the WellSAT 3.0 PEPA scores for each district. The average scores for the PEPA section for the ISD were 19/100 (range: 0–38) for strength and 31/100 for comprehensiveness (range: 13–50). The pilot district strength and comprehensiveness scores were 19/100 and 38/100, respectively. The following PEPA items for the pilot district received a zero: (1) time per week in PE for elementary, middle, and high school, (2) qualifications for PE instructors, (3) professional development for PE instructors, (4) PE substitutions and exemptions, (5) recess for elementary school students, (6) PA breaks during school, and (7) active transportation.

#### 3.1.2. District Self-Evaluation (Step 2)

Table 2 displays the PEPA responses that received a zero during the initial evaluation and the school leaders’ responses during the pilot school district self-evaluation, which took place in March of 2022; school leaders present on the call included the elementary, middle, and high school principals and assistant principals, and superintendent. School leaders identified the following PEPA components that were currently being implemented across the district but were not included in their policy: (1) time per week in PE for elementary, middle, and high school, (2) qualifications for PE instructors, (3) professional development for PE instructors, (4) PE substitutions, (5) recess for elementary school students, (6) PA breaks during school, and (7) active transportation.

### 3.2. Preparation

#### 3.2.1. Tailored Policy Language (Step 3)

Table 2 displays the tailored language added to the pilot district wellness policy, which took place in April 2022. Using the district self-evaluation survey responses, all items that received a zero, except for PE exemptions, were originally evaluated with a score of “0” and were upgraded to a “1” with the tailored policy language. PE exemptions remained a score of “0”. Accordingly, tailored language included added information related to PE time, PE qualifications and professional development, PE substitutions, recess, PA breaks during school, and active transportation. When the PEPA section was rescored with the tailored policy language included, there was a 100% increase in strength (score of 38/100) and 132% increase in comprehensiveness (score of 88/100). The same research team conducted the initial evaluation policy and rescoring.

#### 3.2.2. SWOT Analysis (Step 4)

Figure 2 displays the results from the SWOT analysis led by the research staff at the district/partner workshop in April 2022. District leadership in attendance included the district athletic director, human resources administrator, superintendent, food services director, ISD lead nutrition facilitator, ISD director of instructional services, assistant principals and principals from the elementary, middle, and high school, special education teacher, and the curriculum director.

#### 3.2.3. Feasibility of Tailored Policy (Step 5)

Feasibility of the tailored policy was determined in August of 2022. The district team, consisting of the school district superintendent, food service director, and human resources director shared the tailored policy with the pilot district’s external law firm for review. After considering the law firm’s evaluation, the district leadership discussed the feasibility of the tailored policy and identified three barriers to tailored policy adoption and implementation. Barriers included: (1) collective bargaining agreements, (2) redundancy with existing state laws, and (3) current teacher shortages. Considering those three barriers, school leadership deemed the tailored policy unfeasible; therefore, an alternative strategy for district PA programming was identified: integration of classroom PA into their curriculum guide and school improvement plan. A curriculum guide is a structured document that delineates the philosophy, learning experience, instructional resources, and assessments that comprise a specific educational program. A school improvement plan identifies the academic and priority goals of a school along with strategies and action steps that aim to improve the quality of education students receive. The primary users of curriculum guides are teachers whereas the primary users of school improvement plans are school leadership teams (i.e., administrators, teachers, counselors, and individuals with executive leadership authority). Emphasis in both the curriculum guide and school improvement plan were placed on sustaining classroom PA through the continued implementation of the InPACT intervention, as teacher feedback for this program was overwhelmingly positive and intervention-context fit within the ISD was high [23,24].

### 3.3. Implementation

#### Implementation Timeline (Step 6)

Classroom PA was integrated into the pilot district’s curriculum guide as a “non-negotiable” classroom practice in August 2022 for immediate implementation in the 2022–2023 academic school year. Non-negotiables are clear and specific expectations for compliance within school policies and procedures [25]. Teachers are required to implement two classroom PA breaks per day (five breaks are encouraged). In addition, the district restructured their school improvement plan to reflect the whole child. A whole child approach prioritizes long-term development and success of all children rather than a sole focus on academic achievement [26].

Prior to participating in the policy enhancement process in 2021–2022, the focus of the district school improvement goal was to “provide differentiation in the classroom so that 85% of students meet their growth goal and 65% exceed their growth on the Reading and Math NWEA assessments.” This goal was originally written as a three-year goal and within the first year the pilot district achieved 75% of the students meeting their mathematics goal and 58% meeting their reading goal. However, with both leadership changes at the district level and paradigm shifts at the state level, and InPACT implementation at the school level, the pilot district made a dramatic shift centered on the whole child by creating a systems goal for 2022–2023 in the Spring of 2022. The 2022–2023 goal was revised to emphasize implementing “A cohesive system with multi-tiered layered supports related to academic and personal readiness skills with sustainability over time resulting in overall positive impact trends and a 50% increase in capacity through a district wide coaching model.” InPACT was categorized as a tier 1 universal prevention program.

### 3.4. Sustainment

#### Accountability Measures (Step 7)

Accountability measures that were put into place included teacher evaluations to monitor PA practices within the classroom. Monitoring included observations of classroom practice by an evaluator (i.e., the principal and/or other school administrators) based upon curriculum guide criteria, after which the evaluator and teacher met to discuss observations and set learning goals [27]. In the case of classroom PA, teacher performance was evaluated based on the quantity and quality of PA breaks implemented during the observation period to monitor PA program sustainment. Evaluators’ feedback included: (1) reflective coaching by asking teachers to analyze their own instructional practices related to classroom PA, (2) directive feedback on steps to improve the PA implementation, or (3) engagement with resources and professional development opportunities to modify or enhance PA implementation [28]. The teacher subsequently acted by engaging in professional development, self-directed improvement efforts, or incorporating feedback into instructional practices. That evidence-based evaluation system was guided by the Theory of Action [29].

## 4. Discussion

Locally developed strong and comprehensive policy language is a key driver of district wellness policy implementation [30]. Though schools will often use model-template policies, previous research suggests those types of policies are significantly weaker and less comprehensive with lower policy implementation compared to locally developed policies [12,31]. The literature also demonstrates that effective communication between school partners in the local development of policies further increases the sustainability of PA practices by aligning policy language with practice [32]. In the present study, ISD PEPA policies, which used model-template language, were evaluated as weak and less comprehensive compared to national standards [20] (strength: 19/100 vs. 28/100; comprehensiveness: 31/100 vs. 49/100). Further analysis in the pilot district revealed policy language was not aligned with current PEPA practices as it only included strong language related to PE, but information related to recess, activity breaks, PE substitutions, and after school PA programming was absent, despite schools implementing these programs. Our findings build upon previous work that confirmed a new process is needed to systematically align and enhance district wellness policies.

Our seven-step process, guided by EPIS, sought to promote strong and comprehensive policy adoption and implementation by providing an opportunity for district stakeholders to assess the current implementation of wellness PA policies and practices (step 2), collaborating with school peers (steps 3 and 4), and developing a best-fit action plan for increasing PA sustainment (steps 5, 6, and 7). While developing local policy is more effective to reach PA objectives, it can also be challenging for districts to devise. Using process frameworks, such as EPIS, can provide a systematic and evidence-based process to guide local policy alignment and enhancement. Findings from the current study suggested that co-creating tailored PA policy language with school leadership substantially increased the strength and comprehensiveness of a wellness policy. Alignment of the tailored PA policy with current practices resulted in a 100% increase in strength (score of 38/100) and 132% increase in comprehensiveness (score of 88/100). Yet, implementation barriers were identified, impeding the successful adoption and subsequent implementation of the tailored wellness policy.

Previous research has identified common barriers to PEPA policy implementation such as funding, time, and program support [33,34]. To examine perceptions, barriers, and opportunities related to the development, implementation, and monitoring/evaluation of school wellness policies, Agron et al., conducted online surveys, focus groups, and key informant interviews with over 2900 school administrators who represented 1296 school districts across the United States. Inadequate funding was ranked the top barrier among state association leaders and school board members [33]. Board members also noted that competition with other priorities and mandates; teacher contract restrictions; and not enough time in the curriculum for health, nutrition, and PE were additional barriers [33]. Lastly, the need to educate and gain policy support from key non-staff stakeholders, including students, parents, and the community, were discussed [33].

Our findings are consistent with those of Agron et al. [33] as teacher contract restrictions, time constraints and stakeholder buy-in were identified as barriers to the implementation of our tailored policy. Weaknesses cited by school administrators during the ISD/district SWOT analysis also included lack of curriculum integration and potential low buy-in from parents and older students. In addition, the pilot district’s law firm cautioned that allocating more time to student movement in wellness policies could cause additional strain on both teachers and students. For example, time for lunch is negotiated through collective bargaining and is currently set at 35 min for both teachers and students. Recess for students occurs during lunch time, hence mandating a 20-min recess block would limit lunch time to 15 min. If collective bargaining agreements reduced lunch time to 30 min, an unintended consequence of mandated PA language would reduce lunch time even further, thereby negatively impacting child nutrition. The law firm also advised against creating more precise and redundant PA policy language at the district level, which could lead to future legal conflicts as the district may be subjected to greater scrutiny. Finally, the current teacher shortages across the state and nation provided additional challenges to the feasibility of implementing our tailored policy language as there was no guarantee that districts could comply with a mandate requiring certified teachers to teach or oversee PA programming. Our findings and those of Agron et al. [33] confirmed that tailored implementation strategies are needed to overcome those barriers to policy feasibility and subsequent implementation. More importantly, our findings suggested that engaging law firms and policy agencies (i.e., another important stakeholder group) who are key decision-makers in the process of determining policy feasibility should be included in the policy enhancement process from the outset.

Given the high level of stakeholder engagement achieved within the pilot school district, school leadership members were motivated to find alternative strategies for the sustainment of PA programming. They accomplished this task by integrating PA as a “non-negotiable” school practice into their curriculum guide and refocused their school improvement plan to emphasize a whole child approach to education. Similar to wellness policies, curriculum guides and school improvement plans provide action items to help establish practices that are aligned with district goals [35]. Those documents also help to create accountability measures for the implementation of these practices [35]. A unique feature of curriculum guides is that they provide accountability measures at the implementer level rather than the organization level, as they are used by teachers in day-to-day classroom practice. It is important to note, however, that the accountability measures at the implementer level in the form of teacher observations and evaluations are heavily dependent upon the expertise of the evaluator [36]. Evaluators must have the capacity to observe and meet with teachers regularly; develop trusting relationships with the teachers; and be trained to provide clear, specific, and actionable feedback [29,37,38]. Teachers who are observed regularly and receive frequent high-quality feedback are more likely to improve their instructional practices [39,40]. Hence, additional research is needed to determine the quality and quantity of teacher feedback provided in our pilot school district and its impact on classroom PA sustainment.

Several limitations of the current study should also be noted. First, our study findings cannot be generalized to other school districts within the ISD as this pilot district differed in terms of student demographics (e.g., 90% White vs. 55% White). Moreover, our findings were developed in relation to specific barriers encountered within one pilot district. The alternative strategies presented here may not be applicable to school districts facing different barriers. Second, there is no known research on the effectiveness of adding PA language to curriculum guides and school improvement plans compared to wellness policies. Future research is needed to develop the literature on the fidelity of PA program sustainment through curriculum guides, school improvement plans, regular observation, and teacher feedback. Third, it should be noted that a 12–18 month period of stakeholder engagement prior to the policy evaluation, alignment, and enhancement process occurred, which likely contributed to the strong commitment of the pilot district to sustain InPACT through alternative strategies. Finally, qualitative data to evaluate teacher perceptions of classroom PA integration into curriculum guides were not collected. As such, future research is needed to gain teacher perspective on the barriers and facilitators of integrating PA into curriculum guides and school improvement plans, as opposed to wellness policies.

Several strengths of the current study should be noted. First, our seven-step process was piloted in a socio-economically diverse, rural school district. While our study findings may be unique to this district, our policy process can be applied to a variety of school context. Second, a validated tool (WellSAT 3.0) was used to conduct the policy evaluation and provided quantitative assessments to guide the policy alignment and enhancement process. Third, our process was guided by an evidence-based multiphase implementation science framework, EPIS. While EPIS has been used to guide the implementation of evidence-based practices [15,16], few practitioners have used this systematic approach to guide the evaluation, alignment, and enhancement of local district wellness policies. Finally, our enhancement process enabled the co-creation of a stronger and more comprehensive locally developed PA policy as well as the identification of alternative strategies, which could be applied to other subsections of school district wellness policies.

## 5. Conclusions

PA policies are important for PA program sustainment and can be enhanced through a collaborative process. Yet, many barriers to policy implementation exist, which may impede progress. Alternative strategies to policy enhancement were identified in the present study, which may provide greater accountability and enhanced sustainment of PA programs. Creating flexibility within the policy alignment and enhancement process allowed for the current pilot school district to create feasible accountability measures, which were centered around the implementer rather than the organization. Future research should examine the effectiveness of these alternative strategies (i.e., curriculum guides and school improvement plans) for sustaining PEPA practices in school districts. It is recommended that policymakers continue to identify strategies for overcoming barriers to PA policy implementation to support sustainable PA programming.

## Figures and Tables

**Figure 1 ijerph-20-01791-f001:**
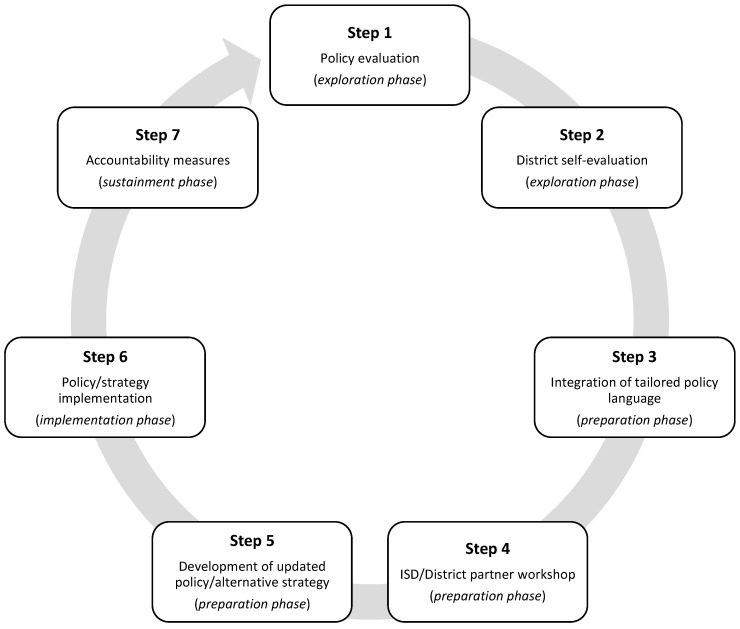
Seven-step policy evaluation, alignment, and enhancement process. ISD, intermediate school district.

**Figure 2 ijerph-20-01791-f002:**
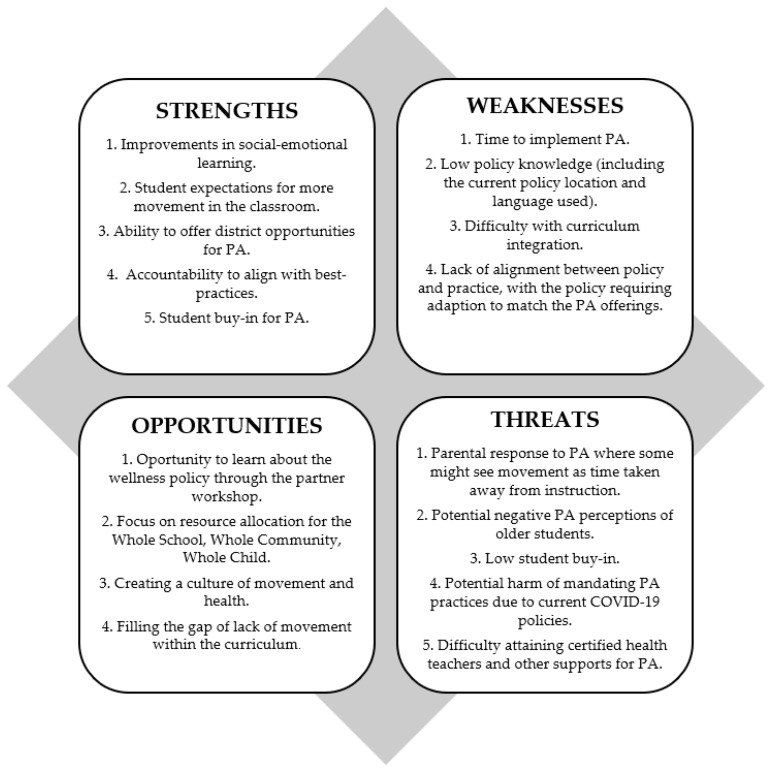
Strengths, weaknesses, opportunities, and threats identified by the school district leaders related to tailored policy implementation, [22].

**Table 1 ijerph-20-01791-t001:** WellSAT 3.0 PEPA scores for each district. Scores have been rounded to whole numbers.

	District 1	District 2	District 3	District 4	District 5	District 6	District 7	District 8	District 9	District 10	District 11	District 12	District 13	District 14	District 15	District 16
**PEPA1:** CSPAP: There is a written PE curriculum for grades K-12.	2	2	1	2	2	2	2	2	2	2	2	2	1	---	---	---
**PEPA2:** The written PE curriculum for each grade is aligned with national and/or state PE standards.	2	2	0	0	2	2	2	2	2	2	2	2	0	---	---	---
**PEPA3:** PE promotes a physically active lifestyle.	2	2	1	2	2	2	2	2	2	2	0	0	2	---	---	---
**PEPA4:** Addresses time per week of PE for all elementary school students.	0	0	0	0	0	0	0	0	0	0	0	0	0	---	---	---
**PEPA5:** Addresses time per week of PE for all middle school students.	0	0	0	0	0	0	0	0	0	0	0	0	0	---	---	---
**PEPA6:** Addresses time per week of PE for all high school students.	0	0	NA*	N/A*	0	0	0	0	0	0	0	0	0	---	---	---
**PEPA7:** Addresses qualifications for PE teachers for grades K-12.	2	2	0	0	2	0	2	0	0	0	0	0	0	---	---	---
**PEPA8:** Addresses providing PE training for PE teachers.	0	0	0	1	0	0	0	0	0	0	0	0	0	---	---	---
**PEPA9:** Addresses PE exemption requirements for all students.	0	0	0	0	0	0	0	0	0	0	0	0	0	---	---	---
**PEPA10:** Addresses PE substitution for all students.	0	0	0	0	0	0	0	0	0	0	0	0	0	---	---	---
**PEPA11:** CSPAP: Addresses family and community engagement in PA opportunities at all schools.	1	0	1	1	1	1	0	0	0	0	1	1	0	---	---	---
**PEPA12:** CSPAP: Addresses before- and after-school PA for all students including clubs, intramural, and interscholastic opportunities.	2	1	1	1	0	1	1	1	0	0	0	0	0	---	---	---
**PEPA13:** Addresses recess for elementary school students.	1	1	1	1	0	0	0	0	0	0	0	0	0	---	---	---
**PEPA14:** CSPAP: Addresses PA breaks for all K-12 students.	0	0	0	0	0	0	0	0	0	0	0	0	0	---	---	---
**PEPA15:** Joint or shared-use agreements for PA participation at all schools.	2	0	0	0	2	1	0	0	0	0	0	0	0	---	---	---
**PEPA16:** Addresses active transport (Safe Routes to School) for all K-12 students who live within walkable/bikeable distance.	0	1	1	0	0	0	0	1	0	0	0	0	0	---	---	---
**Comprehensive Score**	**50**	**44**	**40**	**40**	**38**	**38**	**31**	**31**	**19**	**19**	**19**	**19**	**13**	**N/A**	**N/A**	**N/A**
**Strength Score**	**38**	**25**	**0**	**13**	**31**	**19**	**25**	**19**	**19**	**19**	**13**	**13**	**6**	**N/A**	**N/A**	**N/A**

CSPAP, comprehensive school physical activity programs. PE, physical education, PA, physical activity. N/A, not available. Asterisk denotes districts do not have high schools.

**Table 2 ijerph-20-01791-t002:** Alignment and enhancement of pilot district PEPA policy.

PEPA Item	Question(s)	District Answer	Tailored Language
PEPA 4: Addresses time per week of PE instruction for all elementary school students.	“How much time per week, in MINUTES, of PE instruction do elementary students receive?”	“Elementary students receive instruction for 275 min bi-weekly.”	“Elementary schools should provide 275 min of PE per month.”
PEPA5: Addresses time per week of PE instruction for all middle school students.	“How much time per week, in MINUTES, of PE instruction do middle school students receive?”	“The middle school students receive PE instruction in three ways including an 8-week block for 55 min, for one full trimester in 5th and 6th grade and an elective option in 7th and 8th grade.”	“Middle schools should provide 8 weeks of 55 min per day of PE. Fifth and sixth grade students should participate in 60 days of PE for 55 min per day. Middle school students are encouraged to take one trimester of PE in 7th and 8th grade.”
PEPA6: Addresses time per week of PE instruction for all high school students.	“How much time per week, in MINUTES, of PE instruction do high school students receive?”	“High school students are offered one PE elective of 72 min per trimester of PE instruction.”	“High school students are encouraged to take a one trimester PE elective, for 72 min, 5 days a week.”
PEPA7: Addresses qualifications for PE teachers for grades K-12.	“What, if any, are the qualifications and training for PE teachers for grades K-12?”	“PE teachers are state certified, but no professional development opportunities are offered.”	“PE teachers will be state certified.”
PEPA8: Addresses providing PE training for physical education teachers.	“What, if any, are the qualifications and training for PE teachers for grades K-12?”	“PE teachers are state certified, but no professional development opportunities are offered.”	“All staff involved in PE should be provided with opportunities for professional development.”
PEPA9: Addresses PE exemption requirements for all students.	“What, if any, are the PE exemption and substitution requirements for all students?”	“There are no PE substitutions offered within the (pilot) district, but a varsity letter [sport participation] is an exemption for the class requirement.”	No policy language added**
PEPA10: Addresses PE education substitution for all students.	“What, if any, are the PE exemption and substitution requirements for all students?”	“There are no PE substitutions offered within the (pilot) district, but a varsity letter [sport participation] is an exemption for the class requirement.”	“There will be no substitutions allowed for the PE time requirement.”
PEPA13: Addresses recess for all elementary school students.	“How many MINUTES of recess is provided to all elementary schools?”	“All elementary school students receive 20 min of recess”.	“Schools shall provide at least 20 min of active daily recess to all elementary school students.”
PEPA14: Addresses PA breaks during school.	“What, if any, PA breaks are offered to K-12 students?” “How much time in MINUTES, if at all, is offered for daily PA breaks?” “How much time in MINUTES, if any, are students engaging in daily PA breaks?”	“K-4 students are offered PA breaks through the InPACT program. Grades 5–8 are offered multiple transitions to talk and move through lesson plans or have unstructured GoNoodle breaks. Grades 9–12 are not offered PA breaks.” “10–12 PA minutes are offered, and students spend 10–12 min engaging in PA”.	“Elementary teachers should provide students 10–12 min of daily PA breaks.”
PEPA 16: District addresses active transport (Safe Routes to School) for all K-12 students who live within walkable/bikeable distance.	“Do Safe Route to School plans or programs exist for the school district?”	“Programming does not exist for active transport, but there is the middle school walking club.”	“All students are encouraged to participate in the Walking Club before school hours.”

CSPAP, comprehensive school physical activity programs. PE, physical education. PA, physical activity. N/A, not available. Asterisk denotes districts that do not have high schools.

## Data Availability

The data presented in this study are available on request from the corresponding author. The data are not publicly available due to privacy restrictions.

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
