# Peer review of "A Novel Policy Alignment and Enhancement Process to Improve Sustainment of School-Based Physical Activity Programming"

_ijerph, 2023, doi:10.3390/ijerph20031791_

Round 1

Reviewer 1 Report

The paper is quite interesting, well documented and correctly compiled, and the references are currently update. However, I think it is suitable for publication after clarifying the following aspects:

1.Please see the journal's guidelines for article preparation (IJERPH Microsoft Word template file). These specify that in the abstract you must remove words: Background, Methods, Results, Conclusions.

2. Line 44 – how recently? I recommend that you specify the year to see how recent it is.

3. Line 130 – I recommend that you mention what the 16 items of the PEPA section are, so that the reader can see which items are scored "0" and to know the full content of the section. This evaluation tool is mainly used in the USA, but in Europe it is not known (at least as far as I am concerned) nor accessed (to my knowledge).

4. Results – step 1 - I recommend the tabular introduction of the data collected on the 16 districts so that it can be seen more clearly where the data presented in the text came from.

5. Results – step 3 - The results/16 districts, if they were reproduced, would bring more information to understand the final results.

6. Results – step 6 (lines 303-305) – Was the respective objective achieved or how much of it was achieved? Which were the results? I think it is important to state this, to suggest the probability that the target set for 2022-2023 will be achieved.

7. Results – step 7 – What results were obtained in the evaluation of the teachers?

8. I recommend that the references be reviewed because they do not fully meet the requirements. No DOI was included for all references where it exists.

In the end, I congratulate you for the study you have carried out and I wish you the best in your future research.

Best regards

Author Response

Reviewer 1

  1. The paper is quite interesting, well documented and correctly compiled, and the references are currently update. However, I think it is suitable for publication after clarifying the following aspects.

We appreciate your support of our research efforts, and we hope that we have addressed your concerns adequately in the revised version of this manuscript.

  1. Please see the journal's guidelines for article preparation (IJERPH Microsoft Word template file). These specify that in the abstract you must remove words: Background, Methods, Results, Conclusions.

We have made the suggested change and aligned our abstract with journal guidelines.

  1. Line 44 – how recently? I recommend that you specify the year to see how recent it is.

We have made the suggested change by adding the year in which ESSA was reauthorized (i.e., 2015).

  1. Line 130 – I recommend that you mention what the 16 items of the PEPA section are, so that the reader can see which items are scored "0" and to know the full content of the section. This evaluation tool is mainly used in the USA, but in Europe it is not known (at least as far as I am concerned) nor accessed (to my knowledge).

The 16 items within the PEPA section encompass the following statements: 

(1) CSPAP: There is a written physical education curriculum for grades K-12;

(2) The written physical education curriculum for each grade is aligned with national and/or state physical education standards;

(3) Physical education promotes a physically active lifestyle;

(4) Addresses time per week of physical education for all elementary school students;

(5) Addresses time per week of physical education for all middle school students;

(6) Addresses time per week of physical education for all high school students;

(7) Addresses qualifications for physical education teachers for grades K-12;

(8) Addresses providing physical education training for physical education teachers;

(9) Addresses physical education exemption requirements for all students;

(10) Addresses physical education substitution for all students;

(11) CSPAP: Addresses family and community engagement in physical activity opportunities at all schools;

(12) CSPAP: Addresses before- and after-school physical activity for all students including clubs, intramural, and interscholastic opportunities;

(13) Addresses recess for elementary school students;

(14) CSPAP: Addresses physical activity breaks for all K-12 students;

(15) Joint or shared-use agreements for physical activity participation at all schools; and

(16) Addresses active transport (Safe Routes to School) for all K-12 students who live within walkable/bikeable distance.

These items have been added to the revised Table 1.

  1. Results – step 1 - I recommend the tabular introduction of the data collected on the 16 districts so that it can be seen more clearly where the data presented in the text came from.

We have made the suggested revision and added a table with the results of the PEPA scores for each district. See the revised Table 1.

  1. Results – step 3 - The results/16 districts, if they were reproduced, would bring more information to understand the final results.

We have made the suggested change and added this information to the revised Table 1.

  1. Results – step 6 (lines 303-305) – Was the respective objective achieved or how much of it was achieved? Which were the results? I think it is important to state this, to suggest the probability that the target set for 2022-2023 will be achieved.

The 2021-2022 school improvement goal was originally written as a three year goal. Within the first year the pilot district achieved 75% of the students meeting their mathematics goal and 58% meeting their reading goal. However, with both leadership changes at the district level and paradigm shifts at the State level, the pilot district made a dramatic shift centered on the whole child by creating a systems goal for 2022-2023 in the Spring of 2022. This information is now included in the revised manuscript.

  1. Results – step 7 – What results were obtained in the evaluation of the teachers?

Thank you for this question. Evaluation results are not available yet as the initial implementation began in September 2022. Once evaluation results are available, we plan to publish those findings in a follow-up manuscript.

  1. I recommend that the references be reviewed because they do not fully meet the requirements. No DOI was included for all references where it exists.

We have revised the references in accordance with journal guidelines.

  1. In the end, I congratulate you for the study you have carried out and I wish you the best in your future research.

Thank you again for your support of our research efforts.

Reviewer 2 Report

Researchers have highlighted the importance of policies in school-based physical activity programs. The importance of physical activity for all ages is very well documented. Nevertheless, schools seem to reduce physical activity hours from their everyday schedule. This research suggests a way of promoting school-based physical activity in a very thorough manner. Well done.

Author Response

Reviewer 2

  1. Researchers have highlighted the importance of policies in school-based physical activity programs. The importance of physical activity for all ages is very well documented. Nevertheless, schools seem to reduce physical activity hours from their everyday schedule. This research suggests a way of promoting school-based physical activity in a very thorough manner. Well done.

Thank you for the positive feedback.

Reviewer 3 Report

This is an interesting study examining the strength and comprehensiveness of district wellness policies in a school district and piloting a novel policy alignment and enhancement process for improving sustainment of district-wide physical activity programming.

Authors may consider some issues highlighted below to increase clarity and improve the quality of their manuscript.

Abstract: Authors stated that the “Initial evaluation of the PA policy for the ISD revealed a strength score of 19/100 and 31/100 for comprehensiveness”. A short comment regarding the qualitative evaluation of these scores may also be useful for readers to understand what these scores represent.

Line 76: please explain what PEPA is

The pilot school district consisted of 90.1% White compare to 33% of the ISD. Why this large difference exists? Can this different synthesis of student population to have affect the results of this study? Authors may discuss further this issue.

More information regarding the PE status in the pilot school district may be helpful for understanding the setting of this study. For example, how many hours per week the PE was delivered, if it was delivered by specialist PE teachers or not, the main goals of the PE curriculum etc.

Table 1: why the PEPA items begin with the PEPA 4? What about the first three items?

Table 1: Please explain the difference between District Answer and Tailored Language. Adding a respective note would be helpful.

Line 209: Sustainment is under Step 6: Policy/Strategy Implementation. However, in Figure 1 Sustainment is under Step 7.  Please clarify

Author may consider presenting the results of the SWOT analysis in a 2 × 2 matrix.

Do authors have data to compare and discuss the results of their study in relation to the school level (elementary, middle and high school)? This may be an interesting perspective of the results of this study.

Author Response

Reviewer 3

  1. This is an interesting study examining the strength and comprehensiveness of district wellness policies in a school district and piloting a novel policy alignment and enhancement process for improving sustainment of district-wide physical activity programming.

We appreciate your support of our research efforts.

  1. Authors may consider some issues highlighted below to increase clarity and improve the quality of their manuscript.

We hope that we have addressed your concerns adequately in the revised version of this manuscript.

  1. Abstract: Authors stated that the “Initial evaluation of the PA policy for the ISD revealed a strength score of 19/100 and 31/100 for comprehensiveness”. A short comment regarding the qualitative evaluation of these scores may also be useful for readers to understand what these scores represent.

We have added the following description to the abstract: strength (clear and specific statements that require action) and comprehensiveness (included classroom PA, PE, before-and after-school PA, and family and community engagement).

  1. Line 76: please explain what PEPA is.

We have made the suggested change.

  1. The pilot school district consisted of 90.1% White compared to 33% of the ISD. Why this large difference exists? Can this different synthesis of student population to have affect the results of this study? Authors may discuss further this issue.

We apologize as we incorrectly reported the race/ethnicity demographics for the ISD. Fifty-five percent of students within the ISD self-identify as white, not 33%. We have corrected this error in the manuscript. To your point, we recognize that the difference in demographics between the pilot school and ISD may impede our ability to generalize our findings to other districts within the ISD or across ISDs. This information has been added to the limitations section. Nevertheless, we believe our process, guided by the EPIS framework is generalizable.

  1. More information regarding the PE status in the pilot school district may be helpful for understanding the setting of this study. For example, how many hours per week the PE was delivered, if it was delivered by specialist PE teachers or not, the main goals of the PE curriculum etc.

The revised Table 2 provides the requested information in the column entitled “District Answer.” Elementary students receive PE instruction for 275 minutes bi-weekly. The middle school students receive PE instruction in three ways including an 8-week block for 55 minutes, for one full trimester in 5th and 6th grade and an elective option in 7th and 8th grade. High school students are offered one PE elective of 72 minutes per trimester of PE instruction. PE teachers are state certified, but no professional development opportunities are offered.

  1. Table 1: why the PEPA items begin with the PEPA 4? What about the first three items?

Table 1 (now Table 2) displays the PEPA responses that received a zero during the initial evaluation. PEPA items that received a “1” or “2” were not included in the policy enhancement process. We have added the following statement to provide more clarity: “Table 2 displays the PEPA responses that received a zero during the initial evaluation and the school leaders’ responses during the pilot district self-evaluation.”

  1. Table 1: Please explain the difference between District Answer and Tailored Language. Adding a respective note would be helpful.

District answer refers to the responses from the school district leaders when asked about the PEPA components that received a zero. Tailored Language refers to the wording that was added to the pilot district wellness policy to enhance the score of the policy from “0” to “1”. This information is included in the results section of the manuscript.

  1. Line 209: Sustainment is under Step 6: Policy/Strategy Implementation. However, in Figure 1 Sustainment is under Step 7.  Please clarify.

The word Sustainment on line 209 is italicized to denote a new section has begun. Sustainment is aligned with Step 7.

  1. Author may consider presenting the results of the SWOT analysis in a 2 × 2 matrix.

We have made the suggested change and have presented the SWOT results in a 2 x 2 matrix. See Figure 2.

  1. Do authors have data to compare and discuss the results of their study in relation to the school level (elementary, middle and high school)? This may be an interesting perspective of the results of this study.

Thank you for this suggestion. The InPACT program was only implemented in the elementary school, comparison data are not available. However, the district is currently beginning the process of implementing InPACT in the middle school. As such, we hope to be able to publish those findings in future manuscripts.

Reviewer 4 Report

Frankly speaking, the present study is highly meaningful and valuable, in that the authors have proposed a novel policy alignment and enhancement process to improve sustainment of school-based physical activity programming. Undoubtedly, this would be very beneficial to enhance and strengthen students physical activity in daily life at school, and also be conducive for other different kinds of schools to make feasible, practicable, and reasonable exercise programs based on this program, so that the students physical activity can be improved in a large extent. Given that, I advise this work should be accepted. In addition, only three aspects should be revised and rectified, and the specific suggestions have been listed as follows. Thank you!

Firstly, the abstract should be written by the non-structural form. And then the first letter for all keywords should be shown with lower-case form. Lastly, the references should be also revised and rectified carefully by the corresponding norms of the present journal.

Author Response

Reviewer 4

  1. Frankly speaking, the present study is highly meaningful and valuable, in that the authors have proposed a novel policy alignment and enhancement process to improve sustainment of school-based physical activity programming. Undoubtedly, this would be very beneficial to enhance and strengthen students’ physical activity in daily life at school and be conducive for other different kinds of schools to make feasible, practicable, and reasonable exercise programs based on this program, so that the students’ physical activity can be improved in a large extent. Given that, I advise this work should be accepted. In addition, only three aspects should be revised and rectified, and the specific suggestions have been listed as follows. Thank you!

We appreciate your support of our research efforts, and we hope that we have addressed your concerns adequately in the revised version of this manuscript.

  1. Firstly, the abstract should be written by the non-structural form.

We have made the suggested change.

  1. And then the first letter for all keywords should be shown with lower-case form.

We have made the suggested change.

  1. Lastly, the references should be also revised and rectified carefully by the corresponding norms of the present journal.

We have made the suggested change.